# A Combinatorial Single-Molecule Real-Time and Illumina Sequencing Analysis of Postembryonic Gene Expression in the Asian Citrus Psyllid *Diaphorina citri*

**DOI:** 10.3390/insects15060391

**Published:** 2024-05-28

**Authors:** Qin Zhang, Can Zhang, Hong Zhong, Qing He, Zhao-Ying Xia, Yu Hu, Yu-Xin Liao, Long Yi, Zhan-Jun Lu, Hai-Zhong Yu

**Affiliations:** 1College of Life Sciences, Gannan Normal University, Ganzhou 341000, China; zhangqin0500092@163.com (Q.Z.); zc0502222022@163.com (C.Z.); zh0407292228@163.com (H.Z.); hq0309151111@163.com (Q.H.); xzy20040824@163.com (Z.-Y.X.); wtcs1234@163.com (Y.H.); lyx678253@163.com (Y.-X.L.); yilongswu@163.com (L.Y.); luzhanjun7@139.com (Z.-J.L.); 2National Navel Orange Engineering Research Center, Gannan Normal University, Ganzhou 341000, China

**Keywords:** *Diaphorina citri*, single-molecule real-time sequencing, Illumina RNA sequencing, Hippo signal pathway, RNAi

## Abstract

**Simple Summary:**

*Diaphorina citri* is an important transmission vector of *Candidatus* Liberibacter asiaticus (*C*Las), the causal agent of citrus Huanglongbing (HLB). The control of HLB mainly depends on the management of *D. citri*. The efficiency of *D. citri* nymphs in acquiring *C*Las bacteria is significantly higher than that of the adults, while the transmission efficiency of *C*Las bacteria by adults is significantly higher than that of nymphs. Therefore, it is of great significance to research the functions of key genes during the molting of fifth-instar nymphs. In this study, SMRT and Illumina sequencing were performed on *D. citri* fifth-instar nymphs and adults. SMRT-Seq-generated full-length transcripts provide valuable information for improving functional gene research in *D. citri*. Additionally, Illumina sequencing revealed that the Hippo pathway played an important role in regulating the transition of *D. citri* from fifth-instar nymphs to adults. Our findings provide useful reference information and lay a foundation for controlling *D. citri*.

**Abstract:**

Huanglongbing (HLB) is a systemic plant disease caused by ‘*Candidatus* Liberibacter asiaticus (*C*Las)’ and transmitted by *Diaphorina citri*. *D. citri* acquires the *C*Las bacteria in the nymph stage and transmits it in the adult stage, indicating that molting from the nymph to adult stages is crucial for HLB transmission. However, the available *D. citri* reference genomes are incomplete, and gene function studies have been limited to date. In the current research, PacBio single-molecule real-time (SMRT) and Illumina sequencing were performed to investigate the transcriptome of *D. citri* nymphs and adults. In total, 10,641 full-length, non-redundant transcripts (FLNRTs), 594 alternative splicing (AS) events, 4522 simple sequence repeats (SSRs), 1086 long-coding RNAs (lncRNAs), 281 transcription factors (TFs), and 4459 APA sites were identified. Furthermore, 3746 differentially expressed genes (DEGs) between nymphs and adults were identified, among which 30 DEGs involved in the Hippo signaling pathway were found. Reverse transcription–quantitative PCR (RT-qPCR) further validated the expression levels of 12 DEGs and showed a positive correlation with transcriptome data. Finally, the spatiotemporal expression pattern of genes involved in the Hippo signaling pathway exhibited high expression in the *D. citri* testis, ovary, and egg. Silencing of the *D. citri* transcriptional co-activator (*DcYki*) gene significantly increased *D. citri* mortality and decreased the cumulative molting. Our results provide useful information and a reliable data resource for gene function research of *D. citri*.

## 1. Introduction

The Asian citrus psyllid (*Diaphorina citri*) is the major vector of *Candidatus* Liberibacter asiaticus (*C*Las), the primary agent of huanglongbing (HLB), a serious disease of citrus worldwide [1]. HLB causes tremendous economic loss to the citrus industry every year throughout the world. To date, the control of HLB has mainly relied on planting pathogen-free nursery stock, removing inoculum by destroying *C*Las-infected trees and management of *D. citri* [2]. Currently, the inability to artificially produce *C*Las bacteria has resulted in a lack of effective drugs. Therefore, the control of HLB mainly focuses on vector control [3]. To date, insecticidal applications have been widely adapted for controlling *D. citri*. However, indiscriminate chemical application induces several serious problems, such as environmental pollution, pesticide residue, and insecticide resistance [4]. Therefore, it is critical to explore new, eco-friendly measures to control *D. citri*.

*C*Las bacteria can systematically invade various tissues of *D. citri* in a persistent, propagative manner [5]. During the circulative, propagative transmission cycle of *C*Las, *D. citri* acquires *C*Las bacteria from the infected plants by phloem ingestion. Then, *C*Las travels from leaf veins into the midgut, crossing the midgut epithelial cell layer to circulate in the hemolymph. Finally, the *C*Las bacteria accumulate in salivary glands and circulate through *D. citri* mouthparts [6]. Several studies reported that *D. citri* nymphs acquire *C*Las bacteria much more efficiently than adults, likely because nymphs ingest from the phloem more frequently and for longer durations than adults [7,8,9]. Furthermore, *D. citri* nymphs can only stay on individual trees and cannot migrate to adjacent trees by flight. However, adults can migrate between healthy and *C*Las-infected trees by flying or jumping. Therefore, it is important to target the key genes in process of *D. citri* molting for the control of HLB. In previous research, we identified some key genes involved in chitin metabolism and cuticle synthesis from *D. citri* transcriptome and genome database, including *chitin synthase* (*DcCHS*) and *cuticle protein 64* (*DcCP64*). Silencing of *DcCHS* and *DcCP64* significantly increased *D. citri* mortality and disrupted the molting from the nymph to adult stages, resulting in malformed phenotypes [10]. These results indicated that *DcCHS* and *DcCP64* can be used as the potential targets for controlling *D. citri*.

The transcriptome serves as the intermediary connecting genomic and proteomic data with the biological functions of genes. The transcriptomic analysis of insects provides reliable theoretical support for investigating the prevention and control measures of pests [11]. The transcriptome sequences generated by sequencing platforms through short-read sequencing play an important role in identifying key genes or biological pathways related to growth and development [12,13]. However, the second-generation transcriptome cannot produce full-length sequences due to the limitation of sequencing length [14,15]. In recent years, transcriptome sequencing has been widely used to identify differentially expressed genes (DEGs) in *D. citri*. Yang et al. used RNA-seq to identify novel genes and provide a high-resolution view of the *D. citri* transcriptome throughout development [16]. Additionally, transcriptome sequencing was utilized to screen the response of key genes to *C*Las infection, insecticide resistance, and heat shock [17,18,19]. Unfortunately, DEGs were annotated in these transcriptome sequencing studies based on short-read-based assembly Diaci v1.1 and the *D. citri* v2.0 genome database [20,21]. In addition, the short-length sequencing of full-length transcripts obtained by assembly is incomplete and may result in low-quality transcripts and false annotation [11]. Therefore, it is crucial to improve the transcriptome sequencing depth of *D. citri* and obtain full-length transcript information.

Compared with second-generation sequencing technology, third-generation (single-molecule real-time (SMRT)) sequencing technology presents the following main advantages: long read lengths, efficient analysis of alternative splicing and exon–intron structure, and high consistency and accuracy [22,23]. In recent years, SMRT sequencing combined with Illumina sequencing has been widely utilized for insect gene function research. Feng et al. obtained 10,364 isoforms and identified 259 DEGs by SMRT sequencing in *Odontotermes formosanus* under *Serratia marcescens* treatment conditions [24]. In *Agasicles hygrophila*, 335,045 reads of insert (ROIs) and 158,085 full-length, non-chimeric (FLNC) reads were generated using SMRT sequencing. Additionally, 145 alternative splicing events, 12,753 simple sequence repeats (SSRs), and 16,205 coding sequences were identified [25]. Xu et al. reported the DEGs of *Opisina arenosella* at different developmental stages using Illumina sequencing combined with PacBio single-molecule real-time sequencing, and results showed that 6493 were long noncoding RNAs, and 2510 represented alternative splicing events [11]. These studies revealed that full-length transcriptomic sequencing exhibits significant advantages in obtaining complete transcripts and can be used in conjunction with Illumina sequencing to identify key genes or pathways.

In the current study, SMRT and Illumina sequencing were performed on *D. citri* fifth-instar nymphs and adults. All transcripts obtained from SMRT sequencing were calibrated by Illumina sequencing. The full-length transcripts were annotated, and long noncoding RNAs (lncRNAs), coding sequences (CDSs), SSRs, transcription factors (TFs), and AS events were analyzed. Subsequently, DEGs were identified between fifth-instar nymphs and adults based on Illumina sequencing, and RT-qPCR validated 12 DEGs. Additionally, the spatiotemporal expression pattern of DEGs in the Hippo signaling pathway was also validated. RNAi was performed to knock down the *DcYki* gene and analyze the mortality and the cumulative molting. These results will provide valuable transcriptome resources for *D. citri* gene function research and lay a foundation for controlling *D. citri*.

## 2. Materials and Methods

### 2.1. D. citri Rearing, Sample Collection, and Assay of Weight and Length

*D. citri* specimens were raised using flourishing *Murraya exotica* in a plastic cage with a protective net to prevent adults from escaping. The rearing conditions were set to a constant temperature (27 ± 1 °C), 75 ± 5% relative humidity, and a photoperiod of 12 h: 12 h (L/D) based on the previous method [7]. The fifth-instar nymphs were recognized under a stereomicroscope and collected by a bristle brush. *D. citri* adults were captured from *M. exotica* with a portable aspirator. All obtained samples were kept in 1.5 mL RNase-free centrifuge tubes and kept at −80 °C. Three biological replicates were obtained for each group.

For *D. citri* weight detection, 40 nymphs and adults in each group were weighed using a precision balance. Then, the weight of a single *D. citri* was calculated by the formula (Weight_single_ = Weight_total_/40). The length of *D. citri* was measured using a scale label under a stereoscopic microscope. All measurements involved six biological replicates.

### 2.2. RNA Isolation and cDNA Synthesis

The total RNA was isolated from *D. citri* nymphs and adults using TRIzol reagent (Invitrogen, Carlsbad, CA, USA) based on the manufacturer’s protocol. RNA concentration and purity were detected using a NanoDrop 2000 spectrophotometer (Thermo Fisher Scientific, Waltham, MA, USA). The quality of the total RNA was determined by 1% agarose gel electrophoresis with an Agilent 2100 Bioanalyzer (Agilent Technologies, Palo Alto, CA, USA). The high-quality RNA samples (1.9 ≤ OD_260/280_ ≤ 2.0; OD_260/230_ ≥ 1.9) were used to construct cDNA libraries.

The total RNA was reverse-transcribed with a cDNA synthesis master mix kit (Simgen, Hangzhou, China) according to the manufacturer’s protocol. In brief, a 10 μL reaction mixture containing 2.0 μL of 5×gRNA buffer, 1 μg of total RNA, and moderate RNase-free water was prepared and then incubated at 42 °C for 3 min. Subsequently, 4 μL of 5×RT buffer, 2 μL of RT enzyme mix, and 4 μL of RT primer mix were added to the above mixture and incubated at 42 °C for 15 min and 95 °C for 3 min. The obtained cDNA was preserved at −20 °C for later use.

### 2.3. cDNA Library Construction and Sequencing

The high-quality RNA samples of three replicates were selected from *D. citri* nymphs and adults and then mixed as one sample for PacBio SMRT sequencing. The full-length transcriptome sequencing was performed at Biomarker Technologies (Beijing, China). The full-length cDNAs were synthesized using a SMRTer PCR cDNA synthesis kit (Biomarker, Beijing, China), and then PCR was carried out. The amplified products were purified by 0.8×AMpure PB magnetic beads. Three SMRT cells (1–6 kb) from nymphs and one SMRT cell (1–6 kb) from adults were run on the PacBio RS II platform. The constructed libraries were sequenced by PacBio RS II systems (Pacific Biosciences, Menlo Park, CA, USA).

Illumina sequencing libraries were constructed using Illumina’s NEBNext^®^UltraTM RNA Library Prep Kit (NEB, USA) according to the manufacturer’s instructions. The quality of six libraries was assessed using an Agilent 2100 Bioanalyzer (Agilent, Palo Alto, CA, USA). Subsequently, the Illumina sequencing was performed by the Illumina NovaSeq 6000 platform at Biomarker Technologies Corporation in Beijing.

### 2.4. Functional Annotation of Full-Length, Non-Redundant Transcripts

The full-length, non-redundant transcripts (FLNRTs) were annotated to the following databases using the BLAST alignments: NCBI non-redundant (NR) proteins, SwissProt, eggNOG, Gene Ontology (GO), Clusters of Orthologous Groups of Proteins (COG), Protein family (Pfam), and the Kyoto Encyclopedia of Genes and Genomes (KEGG). The E-value cut-off was set to less than 1.0 × 10^−5^.

### 2.5. Alternative Splicing (AS) Event, Simple Sequence Repeat (SSR), and Open Reading Frame (ORF) Prediction

The GMAP software was used to perform genome mapping and run the alignment program for FLNC sequences [26]. The mapped sequences were further treated by the pbtranscript-ToFU package with min-coverage of 0.85 and min-identity of 0.9. Then, AS events were identified using the AStalavista tool with default parameters [10]. Intron retention indicates that one intron is retained within an exon and simultaneously flanked by two exons. Alternative 5′ or 3′ splicing events indicate that overlapping exons differed at 5′ or 3′ splice junctions, respectively. Exon skipping events indicate exons absent in other isoforms. Mutually exclusive exons mean that one of two exons is retained in mRNAs after splicing. For simple sequence repeat (SSR) analysis, transcripts longer than 500 bp were screened, and then MISA (http://pgrc.ipk-gatersleben.de/misa/misa.html (accessed on 20 January 2023)) was used. The ORFs of transcript sequences were predicted using the TransDecoder v3.0.0 software [22].

### 2.6. Long Noncoding RNA (LncRNA) Prediction and Transcription Factor (TF) Analysis

LncRNAs were predicted using four computational methods, namely, the Coding Potential Calculator (CPC) [27], Coding-Non-Coding Index (CNIC) [25], predictor of long noncoding RNAs and messenger RNAs based on an improved k-mer scheme (PLEK) and Pfam-scan (PFAM) [26]. In addition, lncRNA target genes were predicted by analyzing the correlation between lncRNA and mRNA expression between samples. Transcription factors in *D. citri* were predicted using AnimalTFDB software (Animal Transcription Factor Database v2.0, http://bioinfo.life.hust.edu.cn/AnimalTFDB2/ (accessed on 20 January 2023)) [11].

### 2.7. Identification of DEGs and Functional Enrichment Analysis

Differentially expressed genes (DEGs) between nymph groups and adult groups were identified using the DESeq R package (1.10.1). The resulting *p* values were adjusted using Benjamini and Hochberg’s approach for controlling the false discovery rate. The resulting FDR (false discovery rate) was adjusted using the PPDE (posterior probability of being DE). FDR < 0.05 and |log2(fold change)| ≥ 1 were set as the threshold for significantly differential expression. Gene Ontology (GO) and KEGG enrichment analysis of the DEGs were performed using the GOseq R packages and KOBAS software, respectively.

### 2.8. dsRNA Synthesis and Microinjection

RNAi was performed to analyze the functions of the *Yki* gene in *D. citri* molting. The dsRNA was synthesized using a TranscriptAid T7 High Yield Transcription Kit (Thermo Scientific, Wilmington, DE, USA) according to the corresponding protocol. The specific primers are listed in Appendix A. The prepared dsRNA (ds*Yki* and ds*GFP*) was diluted to 600 ng/mL working solution using RNase-free water, and 0.1% red food dye was added as an indicator. RNAi was conducted based on the previous protocol [7]. In brief, a total of 20 nL of ds*Yki* was injected into fifth-instar *D. citri* nymphs and then transferred onto fresh *M. exotica* seedlings. The same amount of ds*GFP* was injected as a control. The silencing efficiency of the *Yki* gene was determined by reverse transcription–quantitative PCR (RT-qPCR).

### 2.9. Analysis of Expression Levels of DEGs in Different Samples by RT-qPCR

The accuracy of transcriptome data and the expression levels of genes in *D. citri* Hippo signal pathway was validated by RT-qPCR. All primers are presented in Appendix A. Each 10 μL reaction system contained 5 μL of SYBR Ⅱ, 3.5 μL of ddH_2_O, 0.5 μL of each primer, and 0.5 μL of cDNA template according to the manufacturer’s instructions. The CFX384 Real-Time System was used to conduct the PCR reaction. The relative expression levels were analyzed using the 2^−ΔΔCt^ method. The *glyceraldehyde-3-phosphate dehydrogenase* (*GAPDH*) gene was used as a reference gene. There were three biological replicates for each sample, and three technical replicates were performed.

### 2.10. Statistical Analysis

Statistical analyses were conducted with SPSS 11.0 software. Data are expressed as the mean ± standard deviation (SD) from three independent replicates. Significant differences in different samples were analyzed by Duncan’s test with a significance level of 0.05.

## 3. Results

### 3.1. Detection of Weight and Length of D. citri Fifth-Instar Nymphs and Adults

The fifth-instar nymph of *D. citri* undergoes metamorphosis during the molting process, resulting in a substantial change in its morphology (Figure 1A). In the late stage of the fifth-instar nymph, the *D. citri* body color turns black, and the outer cuticle is shed. The results showed that the body weight of the *D. citri* adult was approximately 0.78 mg, and the nymph was approximately 0.38 mg, indicating that the *D. citri* adult weight was approximately twice as much as that of the nymph. However, the body length of the fifth-instar nymph and adult was approximately 3 mm, and there was no significant change in length between nymphs and adults (Figure 1B). These results suggest that the molting of *D. citri* from the fifth-instar nymph to adult stages is accompanied by a doubling of body weight.

### 3.2. PacBio and Illumina Sequencing of D. citri Nymphs and Adults

Three SMRT cells from nymphs and one SMRT cell from an adult were used for PacBio platform sequencing, producing 20.17 and 20.37 G of clean data, respectively. Moreover, 306,224 and 347,641 circular consensus sequences (CCSs) were obtained, and the mean length was 2634 and 1837 from nymphs and adults, respectively. A total of 218,559 and 256,717 full-length, non-chimeric (FLNC) transcripts were identified from nymphs and adults. Additionally, a total of 16,119 and 20,157 consensus isoforms were determined from nymphs and adults by FLNC cluster analysis, respectively. According to the correction of consensus isoforms, 15,992 high-quality isoforms (accuracy) and 126 low-quality isoforms were obtained from the nymph group, and 20,048 high-quality isoforms and 105 low-quality isoforms were obtained from the adult group. Furthermore, the redundant transcripts were removed using the CD-Hit software. A total of 4307 and 6334 full-length, non-redundant transcripts (FLNRTs) were obtained from nymphs and adults, respectively (Table 1). In the OrthoDB database, approximately 30% and 32% of complete BUSCOs were covered from nymphs and adults by isoforms, which confirmed the completeness of the full-length transcriptome in *D. citri* (Appendix A). The eight longest scaffolds of the *D. citri* genome were also selected to compare the isoform density. The results showed that PacBio SMRT sequencing data had higher gene and transcript density than the *D. citri* reference genome (Appendix A). Therefore, the *D. citri* genome was enriched with the PacBio SMRT results and used for further analysis. After Illumina sequencing, 20.17 and 23.97 Gb of clean data were obtained from fifth-instar nymphs and adults, respectively. Q30 and Q20 values were approximately 93% and 97%, respectively, and GC content was over 40% (Appendix A). The base number of clean reads ranged from 6,129,989,232 to 7,235,076,766 (Appendix A). The raw reads were submitted to the sequence read archive of NCBI (BioProject: PRJNA1070823).

### 3.3. Functional Annotation of New Transcripts

The functional annotation of new transcripts was conducted by COG, GO, KEGG, KOG, Pfam, Swiss-Prot, eggNOG, and Nr databases using BLAST software (version 2.2.26) [28]. Results showed that 6489 new transcripts were annotated to eight databases. Of these transcripts, 2391 (36.85%), 2894 (44.60%), 2998 (46.20%), 4175 (64.34%), 4616 (71.14%), 3705 (57.10%), 5285 (81.45%), and 6451 (99.41%) were annotated to COG, GO, KEGG, KOG, Pfam, Swiss-Prot, eggNOG, and Nr databases (Appendix A).

### 3.4. Analysis of SSRs, ORFs, and LncRNAs

A total of 4522 SSRs were identified based on PacBio sequencing. Most SSRs were mono-nucleotide motifs (3446, 76.21%), followed by the di-nucleotide (191, 4.22%), tri-nucleotide (375, 8.29%), tetra-nucleotide (55, 1.22%), and penta-nucleotide (4, 0.09%). In addition, the number of compound SSRs was approximately 405 (8.96%) (Appendix A). Among the detected SSR motifs, the mono-nucleotide motif (198.44 Mb) was the most abundant in density, followed by the compound motif (23.32 Mb), tri-nucleotide motif (21.59 Mb), di-nucleotide motif (10.99 Mb), tetra-nucleotide motif (3.16 Mb), and penta-nucleotide motif (0.23 Mb) (Figure 2A).

The potential ORFs were predicted by TransDecoder software (v3.0.0) based on the corresponding criteria. A total of 5496 complete ORFs were identified, among which the length of ORFs between 100 bp and 500 bp was 3987 (72.54%) (Figure 2B; Appendix A).

Four methods were used to predict the LncRNAs, namely, the Coding Potential Calculator (CPC), Coding-Non-Coding Index (CNIC), coding potential assessment tool (CPAT), and Pfam [27,29,30]. The results showed that 2437, 1429, 3073, and 1899 lncRNAs were predicted by CNCI, CPC, Pfam, and CPAT, respectively. The intersection of these four datasets identified 1124 lncRNAs (Figure 3A). All lncRNAs were divided into four types, and the results revealed that 467, 108, 200, and 311 lncRNAs were distributed into long intergenic noncoding RNA (lincRNA), antisense-lncRNA, intronic-lncRNA, and sense-lncRNA (Figure 3B).

### 3.5. Prediction of TFs, AS Events, and APA

A total of 281 TFs were predicted, and zf-C2H2 was most enriched, with 50 TFs, followed by 46 TFs for ZBTB, 40 TFs for miscellaneous, 23 TFs for HMG, and 21 TFs for TF_bZIP (Figure 4A, Appendix A).

Different types of alternative splicing (AS) events were analyzed by Astalavista software, including the alternative 3′ splice site, alternative 5′ splice site, exon skipping, intron retention, and mutually exclusive exon [31]. The results showed that 307 and 287 AS events were identified from *D. citri* nymphs and adults, respectively. In nymphs, the number of intron retention events was 98 (31.92%), followed by alternative 5′ splice site (68, 22.15%) and alternative 3′ splice (62, 20.2%) events. In adults, the highest number of AS events was recorded for the alternative 5′ splice site (75, 26.13%), followed by intron retention events (70, 24.39%) and the alternative 3′ splice site (58, 20.21%) (Figure 4B).

Polyadenylation of most mRNAs at the 3′ end is necessary for transport, localization, stability and translation of RNA in eukaryote protein biosynthesis. Alternative polyadenylation (APA) was recognized from FLNC reads of nymphs and adults using TAPIS pipeline software [15]. The results revealed that 2070 APA sites from 4307 FLNRTs were obtained in nymphs, and 2389 APA sites from 6334 FLNRTs were obtained in adults (Figure 4C,D; Appendix A). 

### 3.6. Identification of Differentially Expressed Genes (DEGs) and Functional Enrichment Analysis

Data from the PacBio SMRT transcriptome and the Illumina transcriptome were used to analyze the differentially expressed genes (DEGs) between nymphs and adults. The results showed that a total of 3746 DEGs were identified, among which 2152 upregulated and 1594 downregulated DEGs were recorded in adults compared to nymphs (Figure 5A; Appendix A). The hierarchical clustering revealed that DEGs with similar expression patterns were clustered together and showed good repeatability among the three biological replicates (Figure 5B).

In GO analysis, most DEGs were mainly involved in the metabolic process, cellular process, and single-organism process in the biological process (BP) term. Most DEGs were distributed in the cell, cell part, and organelle in the cellular component (CC) term. For the molecular function (MF) term, most DEGs were associated with catalytic activity, binding, and molecular transducer activity (Appendix A; Appendix A). Additionally, in the BP term, 161 DEGs were involved in growth, the developmental process, and reproduction. KEGG analysis suggested that these DEGs were annotated to different biological processes, including cellular processes, environmental information processing, genetic information processing, and metabolism (Appendix A). In cellular processes, a total of 38 and 31 DEGs were involved in the peroxisome and lysosome, respectively. In environmental information processing, 22 DEGs were involved in the Hippo signaling pathway, followed by the Wnt signaling pathway (18) and the Hedgehog signaling pathway (16). In genetic information processing, 18 and 14 DEGs were involved in protein processing in the endoplasmic reticulum and ubiquitin-mediated proteolysis, respectively. In metabolism, most DEGs were mainly associated with purine metabolism (38), carbon metabolism (31), biosynthesis of amino acids (27), and oxidative phosphorylation (26) (Appendix A). Moreover, the DEGs were also annotated to the eggNOG (evolutionary genealogy of genes: non-supervised orthologous groups) database, and the results showed that 329 DEGs were related to signal transduction mechanisms, followed by posttranslational modification, protein turnover, and chaperones (254, 10.37%).

### 3.7. Validation of DEGs by Reverse Transcription–Quantitative PCR (RT-qPCR)

Twelve DEGs were randomly selected for RT-qPCR analysis to validate the transcriptome data. The expression levels of eight DEGs in adults were significantly higher than those in nymphs, namely, *MTL-like protein*, *serine protease esterase-like*, *transferrin*, *circadian clock-controlled protein*, *repetitive proline-rich cell wall protein*, *FAD-linked sulfhydryl oxidase ALR*, *cathepsin L1-like*, and *vitellogenin-1-like* (Figure 6). The other four genes in nymphs were significantly higher than those in adults, namely, *glycine-rich wall structural protein 1.8*, *histone-lysine N-methyltransferase*, *pupal cuticle protein C1B*, and *elastin-like* (Figure 6). These results showed that the expression trend was positively correlated between RT-qPCR and transcriptome data.

### 3.8. Analysis of Spatiotemporal Expression Patterns of DEGs Involved in the Hippo Signaling Pathway

Based on the abovementioned transcriptome database, KEGG enrichment analysis revealed that thirty DEGs were involved in the Hippo signaling pathway. Among them, the expression levels of twenty-six genes in adults were significantly lower than those in fifth-instar nymphs, whereas four genes had high expression in adults compared with nymphs (Figure 7). Furthermore, eight genes were selected for spatiotemporal expression pattern analysis, namely, *DcYki*, *DcSalvador*, *DcI4-3-3*, *DcDmyc*, *DcDs*, *DcEd*, *DcHth*, and *Dcwg*. The results showed that *DcYki*, *DcSalvador*, *DcDmyc*, and *DcEd* had a high expression in the *D. citri* testis and wing. *DcI4-3-3* and *Dcwg* exhibited high expression in the testis and fat body (Figure 8A). For different developmental stages, RT-qPCR analysis showed that the expression level of eight genes in the egg was significantly higher than that in nymphs and adults. Additionally, the *DcYki* expression level had no significant change from the first-instar nymph to the fourth-instar stage. In contrast, it was markedly upregulated from fourth-instar nymph to fifth-instar nymph stages. *DcDmyc* and *DcEd* also exhibited an obvious fluctuation in the fifth instar stage. *DcSalvador* and *DcHth* have no significant difference from the nymph to the adult stages (Figure 8B).

### 3.9. Silencing of DcYki by RNAi Affects the Development of D. citri

The above results showed that the *DcYki* gene was highly expressed in the fifth-instar nymphs. The fifth instar is the key stage for the transition of *D. citri* nymphs to adults. Synthesized dsRNA was injected into fifth-instar nymphs by a micro-injector (Figure 9A). RT-qPCR analysis suggested that the *DcYki* expression level was significantly downregulated at 24 h and 48 h in ds*DcYki*-treated individuals compared with the ds*GFP* treatment group (Figure 9B). Additionally, *D. citri* mortality was significantly higher than that of the control group (ds*GFP* treatment), and cumulative molting was significantly reduced (Figure 9C,D). These results indicated that the *DcYki* gene involved in the Hippo signaling pathway plays an important role in regulating the growth and development of *D. citri*.

## 4. Discussion

The most significant threat of *D. citri* to the citrus industry is mainly HLB transmission. It is critical to control *D. citri* and cut off the transmission path of HLB [2]. Once *D. citri* nymphs develop into adults, the adults can fly, thus accelerating the spread of the HLB between healthy and *C*Las-infected trees. Therefore, it is of great significance to study the functions of key genes during the emergence of fifth-instar nymphs of *D. citri*. We found that the body weight of *D. citri* increased nearly two-fold from fifth-instar nymphs to adults, while the body length of *D. citri* has no significant difference between fifth-instar nymphs and adults. We considered that emerging adults consume much energy from the phloem sap. In our previous research, chitin-metabolism- and cuticle-synthesis-related genes were shown to play an important role in insect molting [1,5,6]. The key genes involved in *D. citri* molting are ideal targets for *D. citri* control.

In recent years, short reads generated by Illumina sequencing have been widely used in RNA-Seq differential gene expression analysis [11]. Due to the limitation of read length, second-generation sequencing cannot generate complete full-length transcripts [32]. A full-length transcriptome based on the PacBio platform can be used to obtain complete transcripts containing the 5′ UTR, 3′ UTR, and polyA tails [33]. Xu et al. obtained 41,938 transcripts by the full-length transcriptome based on PacBio technology [11]. In this study, SMRT-Seq was performed to obtain the full-length transcriptomic information, which was used as reference sequences for the transcriptome analysis of fifth-instar nymphs and adults by Illumina sequencing. A total of 4307 and 6334 full-length, non-redundant transcripts (FLNRTs) were obtained from fifth-instar nymphs and adults, respectively. The average consensus isoform read lengths were 2559 bp and 1778 bp from nymphs and adults, respectively. In previous research, Yang et al. identified 354,726 unigenes with an average length of 925.65 bp across different *D. citri* developmental stages, indicating that SMRT sequencing was superior to Illumina with respect to length [16]. In recent years, the rapid development of genome sequencing technology has promoted the in-depth understanding of vector biology. A high-quality chromosome-level genome of *D. citri* was generated by DNBSEQTM, Oxford Nanopore, and Hi-C technologies. The sequencing results showed that the genome size of *D. citri* is 523.78 Mb, with a scaffold N50 of 47.05 Mb distributed on 13 chromosomes [34]. However, full-length transcript information critical to *D. citri* molting is unclear.

A high annotation ratio reflects the advantages of SMRT-Seq. To obtain comprehensive gene function information, genes were annotated to different databases. We found that 5285 (81.45%) and 6451 (99.41%) transcripts were annotated to eggNOG and Nr databases, respectively. The results indicated a few noncoding sequences, such as lncRNA and intergenic sequences. Additionally, the most full-length sequence annotations were recorded in Pfam (71.14%) and KOG (64.34%), followed by Swiss-Prot annotation (57.10%). Compared with the previous Illumina transcriptome sequencing, the annotation rate of the transcripts is significantly improved [16]. LncRNAs are a kind of non-protein-coding RNA molecule with transcript lengths longer than 200 nucleotides and can engage in various biological processes via epigenetic, transcriptional, and post-transcriptional regulation [35]. In insects, LncRNAs have been shown to play an important role in development [36,37]. In this study, a total of 1080 LncRNAs were identified, namely, 467 lincRNAs, 108 antisense-lncRNAs, 200 intronic-lncRNAs, and 311 sense-lncRNAs. AS events are key elements driving gene and protein expression diversity in eukaryotes [38]. PacBio long-read transcriptome sequencing is superior in identifying AS events [39]. A total of 307 and 287 AS events were identified from *D. citri* nymphs and adults, respectively. The AS events are related to the development of different tissues, such as the brain, liver, and heart [40]. Zhao et al. identified 1804 genes showing AS in each developmental stage and sex, indicating that AS events are ubiquitous in *Plutella xylostella* [41]. In this study, the proportion of intron retention (IR) reached 31.92% in nymphs, while alternative 5′ splice site events accounted for the largest proportion (26.13%) in adults. IR is an alternative splicing mode whereby introns, rather than being spliced out as usual, are retained in mature mRNAs [42]. In our previous research, *D. citri chitin deacetylase 2* (*DcCDA2*) contained two alternative splicing variants, namely, *DcCDA2a* and *DcCDA2b* [43]. We speculated that AS events may play an important role in *D. citri* molting.

Illumina short reads were mapped to PacBio full-length transcripts, and the gene expression levels in fifth-instar nymphs and adults were analyzed. A total of 3746 DEGs were identified by comparing nymphs with adults, namely, 2152 upregulated DEGs and 1594 downregulated DEGs. The transition of fifth-instar nymphs to adults is crucial for the growth and development of *D. citri* and the transmission of HLB [5]. We considered these DEGs to be ideal targets for the control of *D. citri*. Interestingly, KEGG enrichment analysis revealed that most DEGs were mainly enriched into the Hippo signaling pathway. Gene clustering analysis suggested that 26 DEGs involved in the Hippo signaling pathway were downregulated, and 4 DEGs were upregulated in adults compared with nymphs. The Hippo signaling pathway was the first to be found in *Drosophila melanogaster*. This pathway can regulate the development of tissues and organs [44]. In *Bombyx mori*, the Hippo pathway can regulate somatic development and cell proliferation [45]. RT-qPCR analysis indicated that *D. citri* DEGs involved in the Hippo pathway had high expression in the wing, testis, and ovary. Yin et al. found that silencing of the *BmSd* (scalloped) gene by CRISPR/Cas9 induced minute wings in 50% of individuals [46]. In *Drosophila*, the Hippo pathway was involved in regulating follicle stem cell maintenance in the ovary [47]. These results further suggest that the Hippo pathway regulates the development of key tissues, including the wing, testis, and ovary during *D. citri* molting. The transcriptional co-activator Yorkie (Yki), as a member of the Hippo pathway, regulates cell proliferation and apoptosis [48]. In this study, we found that knockdown of *DcYki* significantly increased *D. citri* mortality and decreased cumulative molting. Over the past decade, several studies investigated the biological functions of *Yki* in *Drosophila* and *B. mori*, but its functions in *D. citri* are unclear.

## 5. Conclusions

For the first time, PacBio SMRT-Seq and Illumina sequencing were combined to analyze the *D. citri* transcriptome in fifth-instar nymphs and adults. SMRT-Seq-generated full-length transcripts provide valuable information for improving functional gene research in *D. citri*. Additionally, Illumina sequencing revealed that the Hippo pathway played an important role in regulating the transition of *D. citri* from fifth-instar nymphs to adults. Our findings provide useful reference information and lay a foundation for controlling *D. citri*.

## Figures and Tables

**Figure 1 insects-15-00391-f001:**
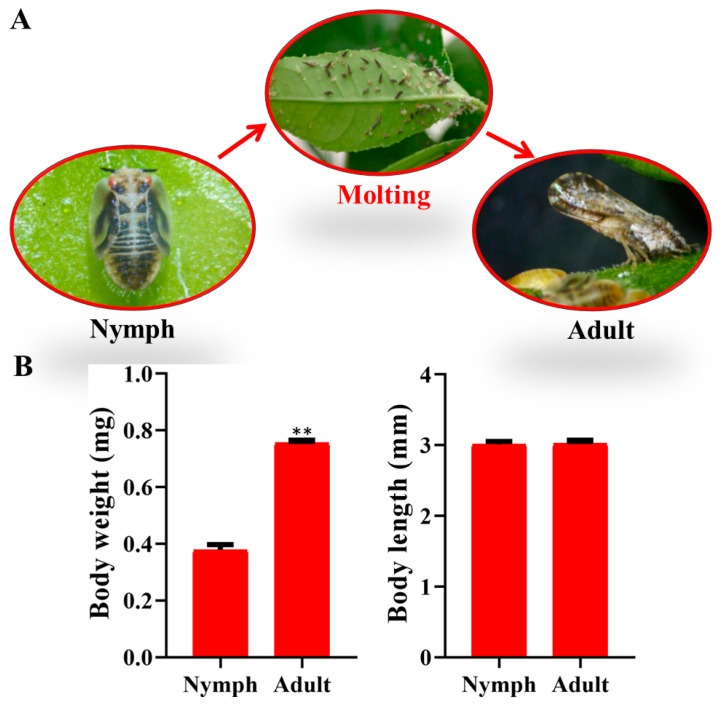
Assay of *D. citri* developmental parameters from fifth-instar nymphs to adults. (**A**) Diagram of development of *D. citri* fifth-instar nymphs into adults. (**B**) Detection of *D. citri* body weight and body length. The left histogram indicates the detection of *D. citri* body weight, and the right histogram indicates the detection of *D. citri* body length. Significant difference is indicated by ** (*p* < 0.01).

**Figure 2 insects-15-00391-f002:**
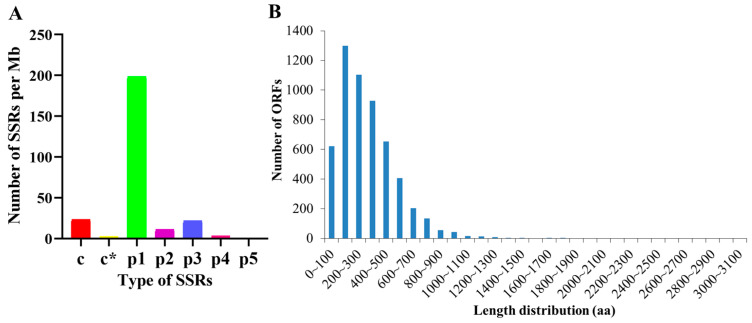
Analysis of SSR density and distribution of open reading frame (ORF) length. (**A**) Analysis of SSR density. c: compound SSR with the common bases; c*: compound SSR without the common bases; p1: mono-nucleotide SSR; p2: di-nucleotide SSR; p3: tri-nucleotide SSR; p4: tetra-nucleotide SSR; p5: penta-nucleotide SSR. (**B**) Distribution of open reading frame (ORF) length.

**Figure 3 insects-15-00391-f003:**
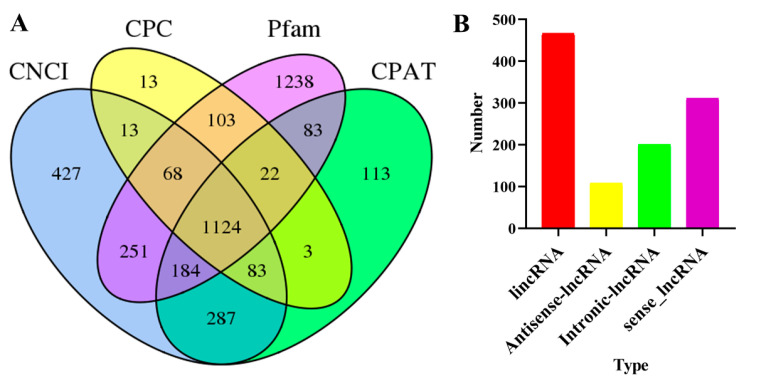
Analysis of *D. citri* lncRNA and transcription factors (TFs). (**A**) Venn graph of lncRNA in *D. citri* from CNIC, CPC, Pfam, and CPAT; (**B**) analysis of different types of lncRNA.

**Figure 4 insects-15-00391-f004:**
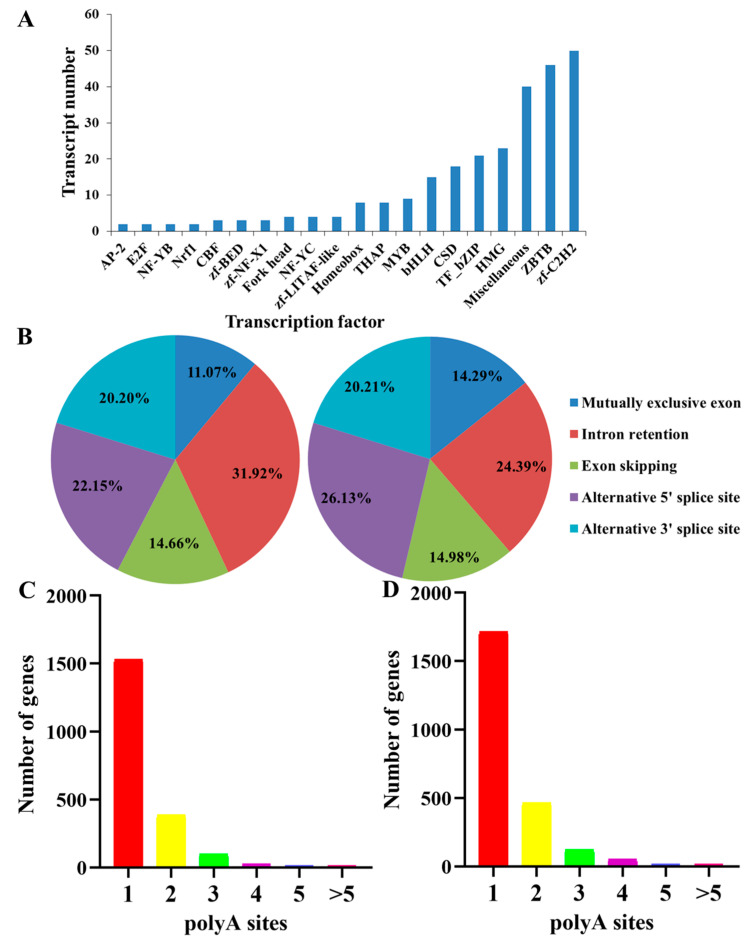
Identification of alternative splicing events and alternative polyadenylation (APA) based on PacBio SMRT sequencing of *D. citri* nymphs and adults. (**A**) Identification of alternative splicing events from *D. citri* nymphs; (**B**) identification of alternative splicing events from *D. citri* adults; (**C**) identification of APA of *D. citri* nymphs; (**D**) identification of APA of *D. citri* adults.

**Figure 5 insects-15-00391-f005:**
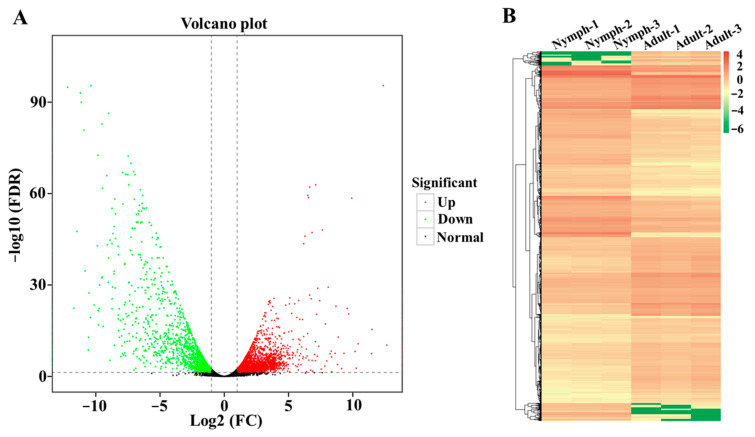
Identification and hierarchical cluster analysis of differentially expressed genes (DEGs) between *D. citri* fifth-instar nymphs and adults. (**A**) The volcano plot of up- and downregulated DEGs in the comparison between *D. citri* fifth-instar nymphs and adults. The scatter diagram shows the results for each gene. The red, green, and black points indicate upregulated genes, downregulated genes, and no difference in expression. (**B**) Hierarchical clustering of DEGs. Columns indicate different samples. Each color box is indicated by the RPKM value of DEGs. Rows represent different DEGs. Red bands indicate a high gene expression level, and green bands indicate a low gene expression level.

**Figure 6 insects-15-00391-f006:**
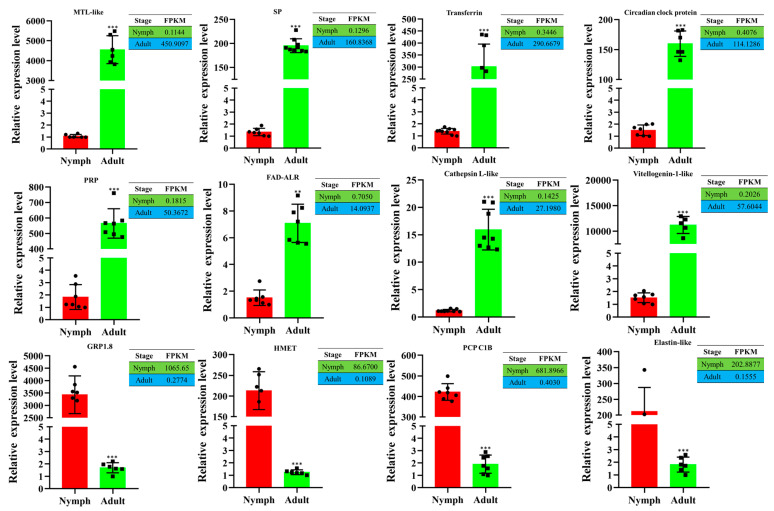
Analysis of expression levels of twelve DEGs between transcriptome and RT-qPCR. The relative expression levels were calculated using the 2^−ΔΔCt^ method. Statistical analysis was performed using the SPSS software. Significant differences are indicated by ** (*p* < 0.01) or *** (*p* < 0.001).

**Figure 7 insects-15-00391-f007:**
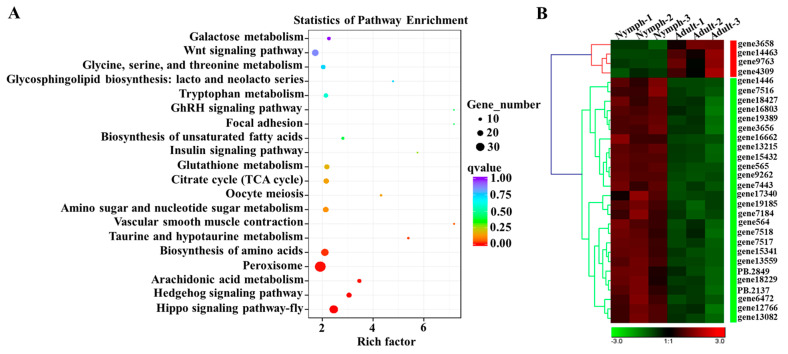
KEGG enrichment analysis of *D. citri* DEGs and hierarchical cluster analysis of genes involved in Hippo signaling pathway. (**A**) KEGG enrichment analysis of *D. citri* DEGs; (**B**) hierarchical cluster analysis of genes involved in Hippo signaling pathway between fifth-instar nymphs and adults.

**Figure 8 insects-15-00391-f008:**
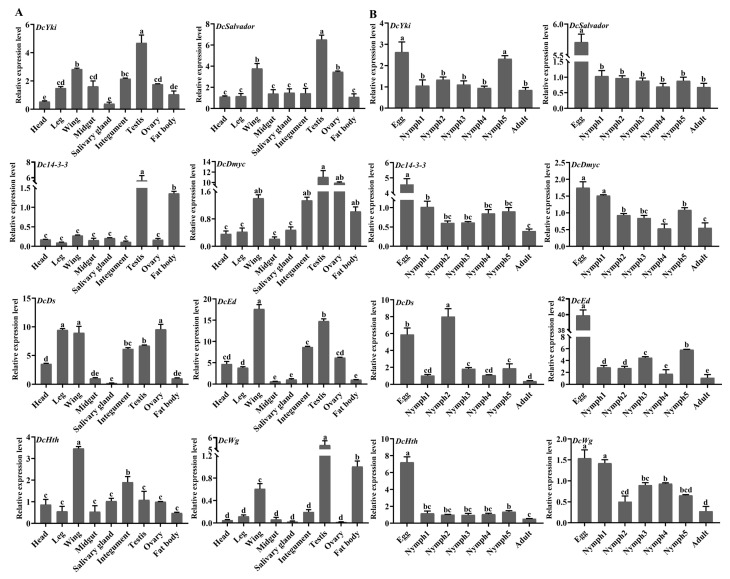
The spatiotemporal expression patterns of eight DEGs involved in the *D. citri* Hippo signaling pathway. (**A**) Relative expression level analysis of eight DEGs in different tissues of *D. citri*, namely, head, leg, wing, midgut, salivary gland, integument, testis, ovary, and fat body. (**B**) Relative expression level analysis of eight DEGs in different developmental stages of *D. citri*. The relative expression levels were calculated using the 2^−ΔΔCt^ method. Statistical analysis was performed using the SPSS software. Different letters indicate significant differences, for example, a, b, c, and d (*p* < 0.05).

**Figure 9 insects-15-00391-f009:**
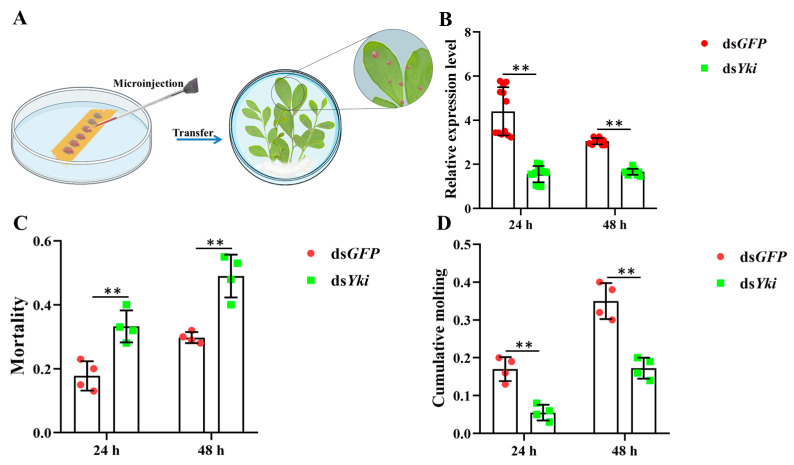
Effect on *D. citri* growth and development after silencing the *DcYki* gene. (**A**) Illustration of the protocol used for RNA interference. (**B**) mRNA level analysis of *DcYki* by RT-qPCR after silencing the *DcYki* gene. The relative expression levels were calculated using the 2^−ΔΔCt^ method. (**C**,**D**) Statistical analysis of *D. citri* mortality and cumulative molting after knockdown of the *DcYki* gene. Statistical analysis was performed using the SPSS software. Significant differences are indicated by ** (*p* < 0.01).

**Table 1 insects-15-00391-t001:** Summary of full-length transcriptome by SMRT-seq of *Diaphorina citri* fifth-instar nymphs and adults.

Data Statistics	Nymph (Number)	Adult (Number)
SMRT cells	3	1
cDNA size	1–6 K	1–6 K
Data size (G)	20.17	23.97
Number of CCSs	306,224	347,641
Read bases of CCSs	806,848,990	638,839,227
Mean read length of CCSs	2634	1837
Mean number of passes	28	44
Number of undesired primer reads	77,099	78,527
Number of filtered short reads	49	89
Number of full-length, non-chimeric reads	218,559	256,717
Full-length, non-chimeric percentage (FLNC %)	71.37%	73.85%
Number of consensus isoforms	16,119	20,157
Average consensus isoform read length	2559	1778
Number of polished, high-quality isoforms	15,992	20,048
Number of polished, low-quality isoforms	126	105
Number of full-length, non-redundant transcripts	4307	6334
Percentage of polished, high-quality isoforms	99.22%	99.48%

## Data Availability

RNA-seq raw reads from this study have been uploaded to NCBI with accession number PRJNA1070823.

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
