# Peer review of "A Combinatorial Single-Molecule Real-Time and Illumina Sequencing Analysis of Postembryonic Gene Expression in the Asian Citrus Psyllid Diaphorina citri"

_insects, 2024, doi:10.3390/insects15060391_

Round 1

Reviewer 1 Report

Comments and Suggestions for Authors

Authors report HLB as the motivation to study the specific stages of D. citri. They used PacBio and Illumina sequencing to study transcriptome of nymph and adult D. citri. The transcriptome results, 30 DEGs involved in Hippo signaling pathway, spatio-temporal expression pattern, and RNAi results are reported. These results are an important contribution to the insect molecular biology field.

The authors report a number of great results in the paper. However, there is a lack of discussion of these results. More work on the discussion part would improve the paper significantly. Most figures also need improvement.

General advice for the authors:

Consider moving some of the figures and tables to supplementary to make the paper more concise and clear. Make sure all the figures are readable, some text on figures is too small. Check text for typing errors as well.

Some specific points:

Line 70: Out of curiosity, in practice, how would these be targeted to control HLB?

Line 260: Report the complete BUSCO output including the gene sets used in the main text. Is 30-32% a good score for D. citri?

Figure 3: The text in figures is hard to read. 3B: There seems to be extra text in y-axis and graph titles

Figure 4: 4A looks good. Text is not readable in 4B and 4C. 4C can be moved to supplementary

Figure 5: Size of the text need to be increased

Figure 6: Legends are barely readable

Figure 9: Not readable

Figure 11: Not readable

Figure 12 is misnumbered

Author Response

  1. The authors report a number of great results in the paper. However, there is a lack of discussion of these results. More work on the discussion part would improve the paper significantly. Most figures also need improvement.

Reply: Thanks for your valuable and thoughtful comments. We have added the detailed discussion for these results. Additionally, we have revised all figures and added to previous manuscript.  

  1. Consider moving some of the figures and tables to supplementary to make the paper more concise and clear. Make sure all the figures are readable, some text on figures is too small. Check text for typing errors as well.

Reply: Thanks for your thoughtful comments. We have moved some of the figures and tables to supplementary materials. Additionally, we also checked text for typing errors carefully.  

  1. Line 70: Out of curiosity, in practice, how would these be targeted to control HLB?

Reply: Thanks for your valuable comments. To date, Diaphorina citri is the only vector of HLB. However, CLas bacteria causing HLB cannot be cultured in artificial medium, and the control of HLB mainly relies on vector control. In previous research, we found that silencing of chitin synthase (DcCHS) and cuticle protein (DcCP64) significantly inhibited molting and increased D. citri mortality (Silencing of the Chitin Synthase Gene Is Lethal to the Asian Citrus Psyllid, Diaphorina citri; Silencing of Chitin-Binding Protein with PYPV-Rich Domain Impairs Cuticle and Wing Development in the Asian Citrus Psyllid, Diaphorina citri). Therefore, these genes were considered as the potential targets for management of HLB.

  1. Line 260: Report the complete BUSCO output including the gene sets used in the main text. Is 30-32% a good score for D. citri?

Reply: Thanks for your thoughtful comments. In this study, approximately 30% and 32% of complete BUSCOs were covered from Diaphorina citri nymph and adult. BUSCO was primarily used to assess the integrity of genome assembly and annotation. Due to the genome assembly of the D. citri is imcomplete, we considered that 30-32% is a relatively good score.    

  1. Figure 3: The text in figures is hard to read. 3B: There seems to be extra text in y-axis and graph titles.

Reply: Thanks for your valuable comments. We have revised the Figure 3 in previous manuscript. 

  1. Figure 4: 4A looks good. Text is not readable in 4B and 4C. 4C can be moved to supplementary.

Reply: Thanks for your thoughtful comments. We have revised the Figure 4 in previous manuscript.

  1. Figure 5: Size of the text need to be increased.

Reply: Thanks for your valuable comments. We have revised the Figure 5 in previous manuscript.

  1. Figure 6: Legends are barely readable.

Reply: Thanks for your thoughtful comments. We have revised the Figure 6 in previous manuscript.

  1. Figure 9: Not readable.

Reply: Thanks for your valuable comments. We have revised the Figure 9 in previous manuscript.

  1. Figure 11: Not readable

Reply: Thanks for your thoughtful comments. We have revised the Figure 11 in previous manuscript.

  1. Figure 12 is misnumbered.

Reply: Thanks for your valuable comments. We have revised the Figure 12 in previous manuscript.

Reviewer 2 Report

Comments and Suggestions for Authors

In the manuscript "The combination of SMRT and Illumina Sequencing Revealed Hippo Signaling Pathway Regulating Diaphorina citri", Zhang et al present a study in which they combined PacBio single-molecule real-time (SMRT) long read sequencing and Illumina short read sequencing to investigate the transcriptome of D. citri fifth-instar nymph and adult.

Their data provide useful information about the functional annotations of the genome. Furthermore, they validated subset of the differential expressed genes and the function of a key gene DcYki in hippo pathway. Overall, the study is well designed, and their conclusion is supported. I have a few minor concerns listed below:

1.     Please add scale bar for the density heatmap in Figure 2

2.     Please make Figure3 legend labels clear, (c, c', p1--5) and (novel.iso.fa...)

3.     Line317, when “lincRNA” is first mentioned, the full name should be noted, too

4.     The text about TFs is in section 3.5 together with AS and APA, while the figure about the TFs is in Figure 4C with lncRNAs. The authors should be consistent with the arrangement of the texts and figures.

5.     In section 3.6 about DEGs, the authors should make it clear about which is the baseline in nymph and adult comparison. For example, up-regulated in adult comparing to nymph, or change term "up-regulated" to "more enriched/abundant in adult".

6.     In Figure 7, The bar for "All gene" and for "DE gene" show similar pattern. What is the difference between these two results? If the result for "All gene" was done with all genes including not differentially expressed ones, why does it show the same result as the "DE gene"?

7.     In section 3.9, besides mortality and molting, does the RNAi affect body weight?

Author Response

  1. Please add scale bar for the density heatmap in Figure 2.

Reply: Thanks for your valuable and thoughtful comments. We have added a scale bar for the density heatmap in Figure 2.

  1. Please make Figure3 legend labels clear, (c, c', p1--5) and (novel.iso.fa...).

Reply: Thanks for your valuable comments. We have revised Figure 3 in previous manuscript.

  1. Line317, when “lincRNA” is first mentioned, the full name should be noted, too.

Reply: Thanks for your thoughtful comments. We have added a full name of lincRNA. 

  1. The text about TFs is in section 3.5 together with AS and APA, while the figure about the TFs is in Figure 4C with lncRNAs. The authors should be consistent with the arrangement of the texts and figures.

Reply: Thanks for your valuable comments. We have revised the Figure 4 in previous manuscript.

  1. In section 3.6 about DEGs, the authors should make it clear about which is the baseline in nymph and adult comparison. For example, up-regulated in adult comparing to nymph, or change term "up-regulated" to "more enriched/abundant in adult".

Reply: Thanks for your thoughtful comments. We have added the related descriptions in previous manuscript.

  1. In Figure 7, The bar for "All gene" and for "DE gene" show similar pattern. What is the difference between these two results? If the result for "All gene" was done with all genes including not differentially expressed ones, why does it show the same result as the "DE gene"?

Reply: Thanks for your thoughtful and valuable comments. Indeed, GO analysis showed that the bar for “All gene” and for “DE gene” show similar pattern. All gene indicates that all genes were identified from transcriptome. DE gene indicates differentially expressed genes between nymph and adult. This figure shows that the gene enrichment of each secondary functions of GO in the DEGs background and all the gene background, reflecting the status of each secondary functions in two backgrounds. The secondary function with obvious proportion difference indicates that the enrichment trend of DEGs at different from that of all genes.   

  1. In section 3.9, besides mortality and molting, does the RNAi affect body weight?

Reply: Thanks for your valuable comments. In previous research, we found that silencing of Yki gene did not affect the weight of D. citri.

Reviewer 3 Report

Comments and Suggestions for Authors

The authors have studied the transcriptomic changes in D. citri at two important developmental stages using a combination of SMRT and Illumina sequencing. Using long read sequencing, they have identified new transcript isoforms, alternate splicing events, SSRs etc. They have also studied differential gene expression at late nymph and adult stages and made observations regarding regulation of Hippo signaling pathway during the molting process. The observation is well supported with qPCR and knockdown experiments.

One of my concern with the study is that the non-redundant transcripts identified using SMRT sequencing are few and account for only 30% BUSCO. The authors should discuss if this is in the expected range or not. They could have attempted de novo assembly with the Illumina reads and compared which of the two assemblies were more complete. On what basis was the DGE analysed with respect to a reference assembly with 30% complete BUSCO. 

Also, several Figures and Tables in the manuscript are redundant or could be included in Supplementary Data. The legends are not descriptive. I have mentioned some of the specific examples and minor queries below.

Table 2 can be provided in Supplementary Data.

Line 281-288: There is duplication in text and table. There is no need for the table if no additional information is provided.

Line 305-307: Can the authors suggest why they are observing >70% of the transcripts below 500 bp? Do they suspect any issues with PacBio sequencing or is this the expected composition in D. citri?

Line 309: Figure 3 legend needs to be more informative.

Line 315-318: There is duplication in text and Figure 4b. There is no additional information in either text or table. One of them can be omitted.

Line 331-334: The authors can provide examples of some genes showing alternate splice events in nymphs and adults and can discuss the importance.

Line 361-366: Legend needs more description. What values are plotted: RPKM or Log Fold change?

Line 462-472 is a repetition of Introduction. It should be kept very brief.

Line 491: Instead of superior, the comparison should be made only with respect to length.

Comments on the Quality of English Language

I have pointed some typographical errors or some sentences that would benefit with rephrasing below

Line 63: frequency should be changed to frequently

Line 108-109: ‘calibrated by illumine sequencing’ needs to be reframed and typos corrected

Line 204: Spelling needs to be corrected D.ciri

Line 217: D. citri to be italicized.

Line 243-245: Figure Legend needs to be rephrased.

Line 302: spelling error ‘scares’

Line 353: Change ‘A Data’ to ‘Data’

Line 355: needs to be rephrased.

Line 417: Four genes had higher expression in adults or nymphs?

Line 507: life activities?

Line 516: ‘sexes”

Line 525: D. citri to be italicized

Line 526, 529, 542: needs to be rephrased

Line 529: Provide full scientific name with italics.

Line 542: to be rephrased

Author Response

  1. One of my concern with the study is that the non-redundant transcripts identified using SMRT sequencing are few and account for only 30% BUSCO. The authors should discuss if this is in the expected range or not. They could have attempted de novo assembly with the Illumina reads and compared which of the two assemblies were more complete. On what basis was the DGE analysed with respect to a reference assembly with 30% complete BUSCO.

Reply: Thanks for your thoughtful comments. In this study, approximately 30% and 32% of complete BUSCOs were covered from Diaphorina citri nymph and adult. BUSCO was primarily used to assess the integrity of genome assembly and annotation. Due to the genome assembly of the D. citri is imcomplete, we considered that 30-32% is a relatively good score.  

  1. Also, several Figures and Tables in the manuscript are redundant or could be included in Supplementary Data. The legends are not descriptive. I have mentioned some of the specific examples and minor queries below.

Reply: Thanks for your valuable comments. We have revised the Figures with unclear legends. Additionally, several Figures and Tables have been added to Supplementary Data.

  1. Table 2 can be provided in Supplementary Data.

Reply: Thanks for your valuable comments. We have added Table 2 to Supplementary data.

  1. Line 281-288: There is duplication in text and table. There is no need for the table if no additional information is provided.

Reply: Thanks for your thoughtful comments. We have added Table 3 to Supplementary data.

  1. Line 305-307: Can the authors suggest why they are observing >70% of the transcripts below 500 bp? Do they suspect any issues with PacBio sequencing or is this the expected composition in D. citri?

Reply: Thanks for your thoughtful comments. In previous analysis of the raw data, we did find that there were too few transcripts below 500 bp. Then, we re-analyzed the sequencing data, but the results were the same as previous data. We considered that the sequenced data is fine. 

  1. Line 309: Figure 3 legend needs to be more informative.

Reply: Thanks for your thoughtful comments. We have added the detailed information for Figure 3 legend.

  1. Line 315-318: There is duplication in text and Figure 4b. There is no additional information in either text or table. One of them can be omitted.

Reply: Thanks for your valuable comments. We have revised the Figure 4B in previous manuscript.

  1. Line 331-334: The authors can provide examples of some genes showing alternate splice events in nymphs and adults and can discuss the importance.

Reply: Thanks for your thoughtful comments. We have added the related descriptions for alternate splice events in discussion.

  1. Line 361-366: Legend needs more description. What values are plotted: RPKM or Log Fold change?

Reply: Thanks for your valuable comments. We have added the related descriptions in previous manuscript.

  1. Line 462-472 is a repetition of Introduction. It should be kept very brief.

Reply: Thanks for your thoughtful comments. We have revised the related descriptions in discussion.

  1. Line 491: Instead of superior, the comparison should be made only with respect to length.

Reply: Thanks for your valuable comments. We have revised the related descriptions in discussion.

  1. Line 63: frequency should be changed to frequently.

Reply: Thanks for your valuable comments. We have revised “frequency” into “frequently” in previous manuscript.

  1. Line 108-109: ‘calibrated by illumine sequencing’ needs to be reframed and typos corrected.

Reply: Thanks for your valuable comments. We have revised “illumine” into “illumina” in previous manuscript.

  1. Line 204: Spelling needs to be corrected D.ciri.

Reply: Thanks for your valuable comments. We have revised “D.ciri” into “D. citri” in previous manuscript.

  1. Line 217: D. citri to be italicized.

Reply: Thanks for your valuable comments. We have revised “D. citri” to italics in previous manuscript.

  1. Line 243-245: Figure Legend needs to be rephrased.

Reply: Thanks for your valuable comments. We have rephrased the Figure legend in previous manuscript.

  1. Line 302: spelling error ‘scares’.

Reply: Thanks for your thoughtful comments. We have revised the spelling error in previous manuscript.

  1. Line 353: Change ‘A Data’ to ‘Data’.

Reply: Thanks for your valuable comments. We have revised “A Data” to “Data” in previous manuscript.

  1. Line 355: needs to be rephrased.

Reply: Thanks for your thoughtful comments. We have revised the incorrect descriptions in previous manuscript

  1. Line 417: Four genes had higher expression in adults or nymphs?

Reply: Thanks for your valuable comments. We have revised the incorrect descriptions in previous manuscript.

  1. Line 507: life activities?

Reply: Thanks for your thoughtful comments. We have revised the incorrect descriptions in previous manuscript.

  1. Line 516: ‘sexes”.

Reply: Thanks for your valuable comments. We have revised “sexe” into “sexes” in previous manuscript.

  1. Line 525: D. citri to be italicized.

Reply: Thanks for your valuable comments. We have revised “D. citri” to italics in previous manuscript.

  1. Line 526, 529, 542: needs to be rephrased.

Reply: Thanks for your thoughtful comments. We have rephrased the related descriptions in previous manuscript.

  1. Line 529: Provide full scientific name with italics.

Reply: Thanks for your valuable comments. We have added a full scientific name with italics in previous manuscript.

  1. Line 542: to be rephrased.

Reply: Thanks for your thoughtful comments. We have revised the related descriptions in previous manuscript.

Round 2

Reviewer 1 Report

Comments and Suggestions for Authors

Authors have sufficiently improved the manuscript

Author Response

Thanks for reviewer's valuable comments. 

Reviewer 3 Report

Comments and Suggestions for Authors

The response from the authors is satisfactory.

Comments on the Quality of English Language

Use of English Language has been improved. There are minor issues, for eg. in Line 287, do you wish to state the number of ORFs?

Author Response

Reply: Thanks for reviewer's valuable comments. We have revised the related descriptions in previous manuscript.